# MGEnrichment: A web application for microglia gene list enrichment analysis

**Justin Jao**, **Annie Vogel Ciernia** *

Department of Biochemistry and Molecular Biology, Djavad Mowafaghian Centre for Brain Health, University of British Columbia, Vancouver, Canada

* annie.ciernia@ubc.ca

## Abstract

Gene expression analysis is becoming increasingly utilized in neuro-immunology research, and there is a growing need for non-programming scientists to be able to analyze their own genomic data. MGEnrichment is a web application developed both to disseminate to the community our curated database of microglia-relevant gene lists, and to allow non-programming scientists to easily conduct statistical enrichment analysis on their gene expression data. Users can upload their own gene IDs to assess the relevance of their expression data against gene lists from other studies. We include example datasets of differentially expressed genes (DEGs) from human postmortem brain samples from Autism Spectrum Disorder (ASD) and matched controls. We demonstrate how MGEnrichment can be used to expand the interpretations of these DEG lists in terms of regulation of microglial gene expression and provide novel insights into how ASD DEGs may be implicated specifically in microglial development, microbiome responses and relationships to other neuropsychiatric disorders. This tool will be particularly useful for those working in microglia, autism spectrum disorders, and neuro-immune activation research. MGEnrichment is available at https://ciernialab.shinyapps.io/MGEnrichmentApp/ and further online documentation and datasets can be found at https://github.com/ciernialab/MGEnrichmentApp. The app is released under the GNU GPLv3 open source license.

**Data Availability Statement:** All relevant data are within the manuscript and its Supporting Information files. Relevant code can be found here: https://github.com/ciernialab/MGEnrichmentApp.

## Author summary

Recent technological and computational advances have produced a massive amount of sequencing data that is often inaccessible to the non-bioinformatician. This is particularly true in multi-disciplinary areas of study such as neuro-immunology, where scientists come from a diversity of background fields. We developed a tool to allow wet-lab scientists without computational skills to utilize previous findings on microglia, the innate immune cells of the brain. Our web hosted tool allows users to compare their genes of interest against a large database of previously published gene lists relevant to microglia and brain disorders. With just a few clicks on the interface, users can upload their genes of interest from mouse or human studies, and query their list by selecting options for running statistical analysis. The application compares the user input to each database list,

**Funding:** This work was supported by the Canadian Institutes for Health Research [CRC-RS 950-232402 to AC]; Natural Sciences and Engineering Research Council of Canada [RGPIN-2019-04450, DGECR-2019-00069 to AC]; Canada Foundation for Innovation / John R. Evans Leaders Fund – Partnerships [CFI 38190 to AC]; SickKids Foundation [NI20-1004 to AC]; and Brain and Behavior Research Foundation [Young Investigator Award 26784 to AC]. This work was supported by resources made available through the NeuroImaging and NeuroComputation Centre at the Djavad Mowafaghian Centre for Brain Health (RRID: SCR_019086). The funders had no role in study design, data collection and analysis, decision to publish, or preparation of the manuscript.

**Competing interests:** The authors have declared that no competing interests exist.

performs a statistical comparison and returns the results to the user, which can be viewed within the application or downloaded for publication. We have included two example datasets of genes from Autism Spectrum Disorder human brain samples. With these example datasets we demonstrate that this type of analysis can be utilized to identify new biological insights and high priority targets for further study in the lab.

This is a *PLOS Computational Biology* Software paper.

## Introduction

With the recent advances in sequencing technology, researchers are increasingly able to generate larger amounts of genomic data. Investigating changes in gene expression has allowed neuroscientists to move beyond the high-level analysis of cellular dynamics, and into the investigation of the molecular and biochemical pathways and networks underlying brain disorders [1]. For example, in the developing brain, early life insults can produce rapid and long-lasting changes to gene expression that alter the neuro-immune system and behaviour [2,3]. Microglia, the brain's resident innate immune cells, appear particularly vulnerable to early life genetic and environmental risk factors for neurodevelopmental, psychiatric and neurodegenerative disorders [4]. As sequencing costs have dropped in recent years [5] and the ability to isolate microglial populations from the brain has expanded, a number of key microglial signature gene lists have been identified across disease models [6] and development [7–10]. The ease at which this data can be generated and incorporated into various experiments has led to gene expression analysis now being utilized not just in hypothesis testing, but also in hypothesis generation [11]. These microglial gene expression differences have been successfully examined across labs and contexts to identify conserved targets and patterns disrupted across brain disorders [2,12]. However, there is currently no central repository for published microglial gene lists, nor a user friendly, non-programmatic interface that allows biologists to statistically test their gene list of interest for enrichment of identified microglial gene lists from other studies.

Several enrichment tools currently exist to assist users in interrogating their gene expression results, such as enrichment of Gene Ontologies using tools such as DAVID [13], Gene Set Enrichment Analysis (GSEA) [14], or KEGG pathways [15,16]. However, these interfaces are not specific to individual cell types nor brain disorders and may not accurately reflect microglial-specific processes or disease states. In comparison, direct gene list comparisons to published microglia datasets can lead to cell type or cell state specific insights into underlying microglial mechanisms. However, this requires access to both a curated database of microglial gene lists and the programmatic skills to implement the analysis and statistics. These obstacles can present a daunting challenge for the non-programming wet-lab scientist. With the increasing use of RNA sequencing (RNAseq) and other expression analysis approaches by biologists, there is a growing need for non-programming based tools that allow for efficient analysis without extensive bioinformatic experience. This need is particularly great in the area of neuro-immunology which attracts researchers from a broad set of backgrounds such as neuroscience, immunology, and others.

Our lab has thus developed MGEnrichment (Microglia Enrichment), a customized web application for performing enrichment testing on a manually curated database of gene lists pertinent to microglia. A key feature of our application is the user's ability to easily upload a

list of genes of interest as either mouse or human gene identifiers, as well as the accessibility of customizing background gene list settings. The application is intended for use by wet-lab scientists who wish to quickly assess the relevance of their gene expression results, and will be of particular interest to those working in the field of microglia research, brain disorders, and neuro-immune activation.

## Design and implementation

The base functionality of the app was built using the R Shiny package (https://shiny.rstudio.com/), and hosted using shinyapps.io by RStudio. MGEnrichment allows the user to upload a list of genes from their experiment in three common gene identifier (ID) formats (Ensembl, Entrez, gene symbols) for either mouse or human (Fig 1). Depending on which gene ID format and species are entered, the database of microglial gene lists (queried from the R biomaRt package [17]) is filtered for the matching ID type and species. Users can select between setting the background as all mouse or human genes, all the genes in the microglial gene list database,

**Fig 1. Model of MGEnrichment.** Users can select either mouse or human gene lists and upload their gene lists of interest either through a CSV file or through entry into the GUI. The input dataset is compared against the database of microglia gene lists to determine enrichment. The GeneOverlap package is used to calculate a one-tailed Fisher's Exact Test for enrichment in each gene list, and FDR corrected p-values are then calculated across all comparisons. The enriched gene results and corresponding statistical significance are then viewable via the GUI, or exportable via CSV. ASD = Autism Spectrum Disorder.

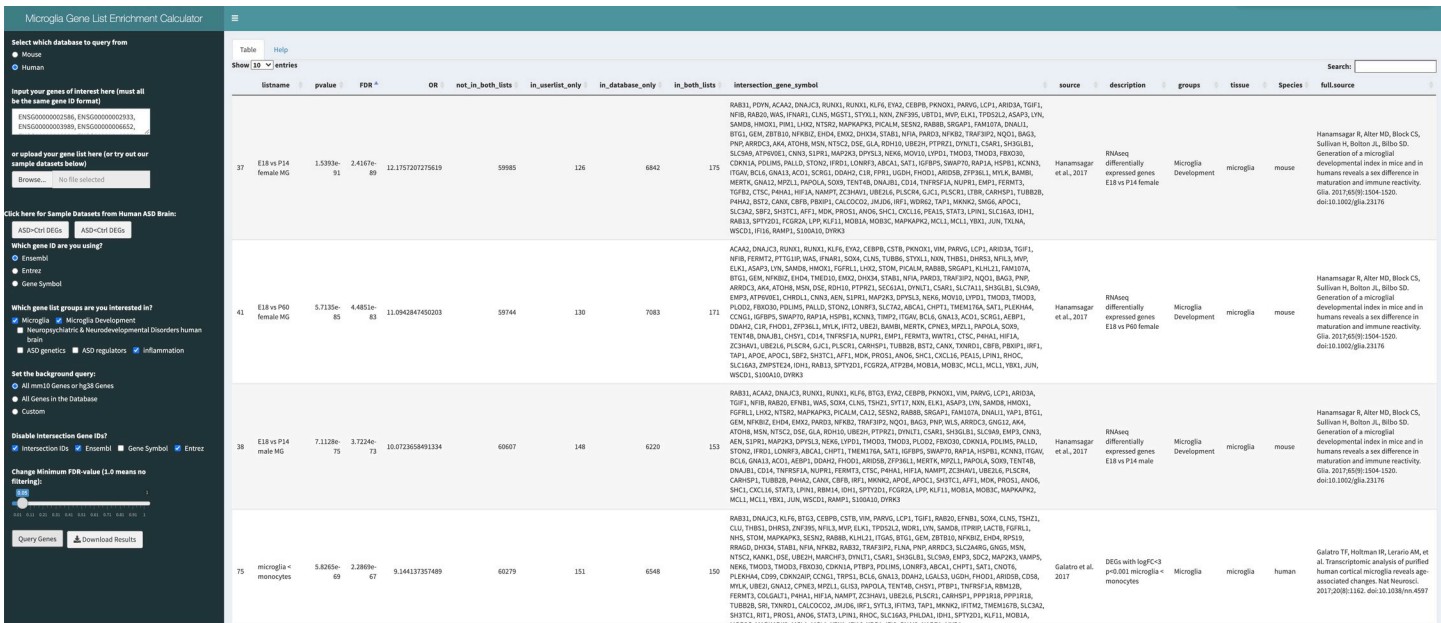

**Fig 2. Preview of MGEnrichment, as previewed on a Web Browser.** The left panel includes user-input and possible modifications to results, while the table on the right outputs the user query results for each gene list.

or an optional user-specified list of background genes. MGEnrichment then performs a one-tailed Fisher's exact test using the GeneOverlap package [18] to compare the overlap between the user's input list and each list in the microglia database. Statistical significance is calculated relative to the background gene list and a False Discovery Rate (FDR) correction is then applied across all comparisons. The level of FDR correction is controlled by the user, allowing for greater flexibility in the statistical threshold used for significance determination.

Enrichment results display several key output variables including the odds ratio, p-value, FDR corrected p-value, and the number and IDs of the overlapping genes for each database list (Fig 2). Information is also provided regarding individual microglial database gene lists, including the group they belong to, a description of the gene list, the species the gene list was collected from, as well as a literature source for where the gene list originates. These results may be viewed directly on the web browser or as a downloaded CSV file.

The database contains 214 unique microglial gene lists from 42 publications pulled from the microglial literature (S1 Table). Gene lists from mouse, rat and human are included. For the mouse version of the database, all human and rat gene IDs were converted to mouse using bioMart. For the human version of the database all mouse and rat gene IDs were converted to human gene IDs using bioMart. The database of gene lists was manually curated from previous literature and includes a wide assortment of microglial relevant gene lists collected from multiple treatments, disease states and developmental timepoints in microglia or brain. Additional gene lists can be added to the database directly using example code provided in the GitHub repository or by request through GitHub. Users may select from several background options including uploading a custom background list. Users can select subsets of gene lists from the database based on six different list categories (groups). Group options include Microglia, Microglia Development, Neuropsychiatric & Neurodevelopmental Disorders, Autism genetics, Autism regulators, and Inflammation. The user can select the groups to be included, allowing for more targeted analysis to a specific subgrouping within the database.

To demonstrate the utility of our approach we created two "toy" datasets that examine gene regulation in ASD. Microglial dysregulation has been observed in ASD postmortem brain samples in terms of altered cellular morphology and gene expression. Specifically, there have been three large-scale, recent RNAseq studies examining differentially expressed genes from human ASD postmortem brain compared to matched controls [19–21]. All three identified immune, and specifically microglial, gene expression as altered in ASD brain [19–21]. We took the published gene lists from these papers, divided them into genes with either increased or decreased expression in ASD and then overlapped the three sets to identify genes consistently identified in at least 2 out of the 3 publications. Users can access these datasets by clicking their respective buttons on the application with the desired species selected, and querying the database to look for gene list enrichments. Alternatively, a compiled supplemental excel spreadsheet (S2 Table) of both mouse and human toy datasets and the corresponding MGErichment results can be downloaded from the GitHub repository. Human enrichments were calculated using Human Ensembl gene IDs, with the background set to "All hg38 Genes", queried against all gene list groups, and with FDR filtering for q<0.05. Similarly, mouse enrichments were calculated using Mouse Ensembl gene IDs (after conversion from human using BioMart), with the background set to "All mm10 Genes", queried against all gene list groups, and with FDR filtering for q<0.05.

## Results

The MGEnrichment app is setup so that users can easily query the microglia database to analyze the gene expression profiles of their lists (mouse or human) compared to selected lists from the database. The provided toy ASD increased gene expression dataset (ASD>CTRL DEGs) produces numerous significant (FDR q<0.05) enrichments with database gene lists. For example, using the Human Ensembl IDs, ASD>CTRL DEGs are significantly enriched for genes with increased expression in schizophrenia, a relationship previously identified [21]. There were also significant enrichments with gene lists important for microglial development, gene regulation (*Sall1* and *Mef2c*) and immune activation (PolyI:C and LPS treatments) (S2 Table). There were also significant enrichments with gene lists generated from microglia from germ free mice, supporting a recent growing literature on the role of the microbiome in ASD [22] and suggesting microbiome disturbances associated with the disorder may contribute to altered brain microglia. From these enrichments, individual genes of interest can be identified among the shared genes to identify novel targets for further investigation. For example, our target toy list (ASD>CTRL DEGs) shares genes with higher expression in amoeboid compared to ramified microglia. These genes might be reasonable targets for further exploration to explain previously observed changes in microglia morphology in human post-mortem brain from ASD [23,24].

Similarly, using the Human Ensembl IDs for genes with decreased expression in ASD brain (ASD<CTRL DEGs) produces significant overlaps with lists for other human neuropsychiatric disorders as well as genes regulated in microglial development (S2 Table). The developmental list enrichments all center around lists of differentially expressed genes between embryonic day 18 (E18) microglia and postnatal microglia (P4, P14 and P60), suggesting that genes with disruption in ASD may impact embryonic microglia maturation towards a postnatal transcriptome.

Together, our two example datasets demonstrate the utility of MGEnrichment in exploring microglial gene regulation in neurodevelopmental disorders. The app can provide both novel insights into differentially expressed gene lists, as well as identification of microglial target genes for further examination.

## Availability and future directions

The code for the application is freely available on our GitHub repository, and released under the GNU General Public License version 3 (GPLv3). By releasing this under an open source license, we aim to provide transparency as to how our program was designed, as well as invite collaboration and contributions from others in the field. Documentation for MGEnrichment is provided within a "help" tab of the web application and at https://github.com/ciernialab/MGEnrichmentApp. All source code is included on the GitHub repository, including the microglia gene list database and instructions for adding in new custom gene lists to the database.

MGEnrichment allows for a targeted approach to understanding microglial biology by leveraging known changes in gene expression across different disease and developmental states. As genomics becomes increasingly intertwined with neuro-immunology and behavioural neuroscience research, the ability to interpret gene expression results within the broader context of microglial biology will be a key skillset for many researchers. We have developed MGEnrichment to accomplish two main goals: firstly, to disseminate an easy to access database of curated microglia-relevant gene lists; secondly, to provide a user-friendly interface for non-programmers to examine their gene lists of interest for impacts on microglial biology. MGEnrichment's hosting on the web through the R Shiny platform allows any user to easily query their gene list of interest and download their results for further analysis.

Future directions for the project include expansion to include additional types of data visualization, such as dot plots to better visualize the level of gene enrichment and network visualizations to support more systems-based analyses. It is our hope that this app will act as a useful tool to bridge the gap between wet and dry-lab scientists in microglial research, and to help traditional behavioural neuroscientists and immunologists to interpret changes in microglial gene regulation.

## Supporting information

**S1 Table. MG Database.** Sheet 1: MG Mouse Database. Includes an entry for each gene list in the curated mouse database, description of the gene list, source/citation, group assignment, species of the original study, tissue and the number of Ensembl mouse IDs within that list. Sheet 2: MG Human Database. Includes an entry for each gene list in the curated human database, description of the gene list, source/citation, group assignment, species of the original study, tissue and the number of Ensembl human IDs within that list.
(XLSX)

**S2 Table. Toy Dataset.** Sheet 1: ASD>CTRL_DEGs_Datasets. Includes the input dataset containing the human and mouse Ensembl IDs for genes identified across 2 out of 3 human brain RNA-seq studies comparing brain samples from ASD and Controls. DEGs show higher expression in ASD compared to Control samples. Human IDs were converted to mouse IDs using BioMart. Sheet 2: ASD<CTRL_DEGs_Datasets. Includes the input dataset containing the human and mouse Ensembl IDs for genes identified across 2 out of 3 human brain RNA-seq studies comparing brain samples from ASD and Controls. DEGs show lower expression in ASD compared to Control samples. Human IDs were converted to mouse IDs using BioMart. Sheet 3: Human ASD>CTRL Results. FDR filtered (q<0.05) enrichment results are shown for all significant enrichments between ASD>CTRL DEGs and gene lists in the MGEnrichment database run using human Ensembl IDs from Sheet 1. Sheet 4: Human ASD<CTRL Results. FDR filtered (q<0.05) enrichment results are shown for all significant enrichments between ASD<CTRL DEGs and gene lists in the MGEnrichment database run using human Ensembl

IDs from Sheet 2. Sheet 5: Mouse ASD>CTRL Results. FDR filtered (q<0.05) enrichment results are shown for all significant enrichments between ASD>CTRL DEGs and gene lists in the MGEnrichment database run using mouse Ensembl IDs from Sheet 1. Sheet 6: Mouse ASD<CTRL Results. FDR filtered (q<0.05) enrichment results are shown for all significant enrichments between ASD<CTRL DEGs and gene lists in the MGEnrichment database run using mouse Ensembl IDs from Sheet 2.
(XLSX)

## Acknowledgments

We would like to thank members of the Ciernia, Tropini and Osborne labs at UBC for helpful feedback on this project.

## Author Contributions

**Conceptualization:** Annie Vogel Ciernia.

**Data curation:** Annie Vogel Ciernia.

**Formal analysis:** Justin Jao, Annie Vogel Ciernia.

**Funding acquisition:** Annie Vogel Ciernia.

**Methodology:** Justin Jao, Annie Vogel Ciernia.

**Project administration:** Annie Vogel Ciernia.

**Resources:** Justin Jao, Annie Vogel Ciernia.

**Software:** Justin Jao, Annie Vogel Ciernia.

**Supervision:** Annie Vogel Ciernia.

**Validation:** Justin Jao.

**Visualization:** Justin Jao.

**Writing – original draft:** Justin Jao, Annie Vogel Ciernia.

**Writing – review & editing:** Justin Jao, Annie Vogel Ciernia.

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
