## [Decision Letter · Decision Letter 0]

17 Jul 2021

Dear Dr. Ciernia,

Thank you very much for submitting your manuscript "MGEnrichment: a web application for microglia gene list enrichment analysis" for consideration at PLOS Computational Biology.

As with all papers reviewed by the journal, your manuscript was reviewed by members of the editorial board and by several independent reviewers. In light of the reviews (below this email), we would like to invite the resubmission of a significantly-revised version that takes into account the reviewers' comments.

We cannot make any decision about publication until we have seen the revised manuscript and your response to the reviewers' comments. Your revised manuscript is also likely to be sent to reviewers for further evaluation.

Sincerely,

Mihaela Pertea

Software Editor

PLOS Computational Biology

Mihaela Pertea

Software Editor

PLOS Computational Biology

Reviewer's Responses to Questions

**Comments to the Authors:**

Reviewer #1: This manuscript describes a novel computational software/tool to assess overlap (and statistical significance of the overlap) between a user’s list of genes of interest and target ensemble of genes expressed by microglia in defined conditions. Microglial cells are at the center of intense research in neuroscience, owing to their roles in neurodevelopmental disorders, neurodegenerative disorders, and basic brain biology in general. As also pointed out by the authors, there has been a tremendous accumulation of high-quality transcriptomic data associated with microglia over these past 5-6 years. These include both bulk RNA-seq and single-cell RNA-seq. Furthermore, many groups now rely on direct comparisons with these published datasets to guide their studies. This is done to either cross-validate their own experimental results or to justify molecular targets of interest for specific experiments.

Given the above, release of MGEnrichment is timely and will likely facilitate data analysis and interpretation. Based on my testing, it is quite user-friendly, and results are quickly generated. Thus, its intended audience (i.e., people not well-versed in computational analyses) should find it accessible. That being said, it is a little perplexing to see that recent seminal work on human microglia were not included in the first iteration of the software. For example, work from Bart Eggen’s lab (PMID 28671693 and 32732419) and Chris Glass’ lab (e.g., PMID 28546318) have provided high-quality, comprehensive datasets of microglial gene expression from human microglia, and these are frequently used as baseline to generate and refine protocols for the generation of ips-derived microglia. My guess is that this may be an issue of setting-up a “human-based” analysis arm so that human genes and their identifiers are properly treated when entered as input. Irrespective of the reasons however, I would be inclined to ask that the authors add a human component, as the resulting enhanced scope would be better aligned with the goal of PLOS Computational Biology. Moreover, given the significant efforts devoted to research on human microglia and ips-derived microglia in the context of neurodegenerative disorders, a human component would necessarily reach a larger audience, which would allow MGEnrichment to set itself better as a key reference in the field.

Reviewer #2: Summary of MGEnrichment: a web application for microglia gene list enrichment analysis

The authors address the lack of a central repository for microglial gene expression signatures by creating MGEnrichment, a non-programmatic web application for microglia gene enrichment analysis. The database includes 166 microglial gene lists from human, mouse and rat. This could assist non-programming scientists in their analyses of gene expression data in a cell-type-specific or disorder-specific way.

This tool is created using the Shiny R package, which allows its users to upload their own gene list and define a background gene list. Using the GeneOverlap R package, MGEnrichment then performs a one-tailed Fisher’s exact test to compare the input data against each of the lists from the database. A combination of any subset or all lists from the microglia database can be used for the enrichment analysis. The subsets include: microglia, microglia development, neuropsychiatric & neurodevelopmental disorders human brain, autism genetics, autism regulators, and inflammation. The p-value is then computed to evaluate statistical significance, followed by a False Discovery Rate (FDR) calculation. The app provides users the flexibility to set an FDR threshold. Finally, the output is presented on the web browser and a downloadable CSV file. This could be particularly useful for scientists with an interest in microglia and neuro-immunology.

To validate the applicability of MGEnrichment, Jao et al used two “toy” datasets: ASD>CTRL DEGs and ASD<ctrl asd="" degs.="">CTRL DEGs is a gene list that is increasingly expressed in autism spectrum disorder (ASD), whereas ASD<ctrl an="" asd="" brain.="" come="" dataset.="" datasets="" decreased="" degs="" examined="" expression="" four="" from="" gene="" in="" is="" postmortem="" querying="" rnaseq="" studies="" that="" the="" these="" upon="">

In summary, we found the article and web tool of high quality and of appropriate scope for PLoS Comp Bio. We (myself and members of my lab) were able to adequately test the tool and found its functionality helpful in interpreting microglial gene sets that we had generated in our lab. We were able to use the app to understand the enrichment of multiple gene sets related to microglia. The careful curation and standardization of multiple microglia-related gene sets is an especially valuable contribution of this work.

We had a few minor comments with the manuscript / list of datasets with microglial datasets:

• It’d be helpful to include a table in the manuscript listing the datasets and publications used

• How carefully was the literature curated, in particular, the list of human microglial subtypes? We found the omission of the datasets presented in Olah et al., 2020 as one such missing dataset we had expected to be included: https://www.nature.com/articles/s41467-020-19737-2

In using the associated web application, we noticed a few points for improvement that would greatly increase the usability of the application.

• Given that our gene lists were based on analyses of human cells, we found it inconvenient to have to first convert these genes to the equivalent gene names in the mouse. This seemed non-intuitive and unnecessary, especially as a number of the gene lists are based on human gene expression profiles. I suggest adding functionality to allow the user to enter what species their gene names are entered as, and do the conversion of the symbols within the app.

• The gene list groups that are selected by default are non-intuitive. Given that this is a microglia enrichment tool, it seems more useful and intuitive to have only the microglia, microglia development, and inflammation gene list groups selected by default.

• It’d be more useful to pick a more meaningful default for the minimum FDR value – I’d suggest a value of 0.1 instead of 1.0.

• The notAnotB inAnotB terminology is not intuitive. Consider using the terms N K n k from hypergeometric test: https://en.wikipedia.org/wiki/Hypergeometric_distribution

• There are a lot of columns displayed within the tool – are all of these really necessary? I don’t find it particularly helpful to show both the intersection IDs, intersection ensembl, intersection mgi symbol, etc etc. I think merely showing the gene symbols would be more than sufficient. Showing all of these columns makes each row within the app very long and challenging to scroll through.

• The sorting of results that are displayed in the table would be considerably more intuitive if the table was sorted by default in order of increasing pvalue or FDR.

• Is it possible to have the source publications be listed as hyperlinks with links to the publications at pubmed?

• Please think carefully about where columns in the table should appear. For example, I found the shortnames for each condition somewhat difficult to parse. The problem would likely be fixed if the columns listname, shortname, and description were closer to one another (also – why do you need all 3).</ctrl></ctrl>

**Have the authors made all data and (if applicable) computational code underlying the findings in their manuscript fully available?**

Reviewer #1: Yes

Reviewer #2: Yes

PLOS authors have the option to publish the peer review history of their article (what does this mean?). If published, this will include your full peer review and any attached files.

Reviewer #1: No

Reviewer #2: No
---

## [Decision Letter · Decision Letter 1]

19 Oct 2021

Dear Dr. Ciernia,

We are pleased to inform you that your manuscript 'MGEnrichment: a web application for microglia gene list enrichment analysis' has been provisionally accepted for publication in PLOS Computational Biology.

Best regards,

Mihaela Pertea

Software Editor

PLOS Computational Biology

Mihaela Pertea

Software Editor

PLOS Computational Biology

Reviewer's Responses to Questions

**Comments to the Authors:**

Reviewer #1: The authors have addressed my original concerns with this resubmission.

Reviewer #2: No further comments. The authors have done an excellent job responding to my prior review. I especially commend the authors on implementing all of the suggestions to improve the usability of the web app.

**Have the authors made all data and (if applicable) computational code underlying the findings in their manuscript fully available?**

Reviewer #1: Yes

Reviewer #2: Yes

PLOS authors have the option to publish the peer review history of their article (what does this mean?). If published, this will include your full peer review and any attached files.

Reviewer #1: No

Reviewer #2: **Yes: **Shreejoy J Tripathy

---

## [Editor Report · Acceptance letter]

1 Nov 2021

PCOMPBIOL-D-21-01028R1 

MGEnrichment: a web application for microglia gene list enrichment analysis

Dear Dr Ciernia,

I am pleased to inform you that your manuscript has been formally accepted for publication in PLOS Computational Biology. Your manuscript is now with our production department and you will be notified of the publication date in due course.

With kind regards,

Livia Horvath
